# TEXT-DRIVEN IMAGE EDITING USING CYCLE-CONSISTENCY-DRIVEN METRIC LEARNING

## ABSTRACT

We present a simple but effective training-free approach for text-driven image-to-image translation based on a pretrained text-to-image diffusion model. Since a naïve application of the pretrained diffusion models for image editing often significantly degrades the structure or background in the source image, we revise the original backward process for generating target images to properly align with the target task while preserving the background or structure of a source image. To this end, we derive a new guidance objective term based on a combination of two objectives: maximizing the similarity to the target prompt rather than the source prompt based on the CLIP score and minimizing the distance to the source latent variables. Moreover, unlike existing methods based on diffusion models, we exploit the cycle-consistency objective to preserve the background of the source image, for which we iteratively alternate the optimization procedure for the source and target latent variables. Experimental results demonstrate that the proposed method achieves outstanding image-to-image translation performance on various tasks when combined with the pretrained Stable Diffusion model.

## 1 INTRODUCTION

The main objective of the text-driven image editing task is to translate a given source image into the target domain while preserving the overall structure or background in the source image. Generative Adversarial Network (GAN) (Goodfellow et al., 2014) has been widely used to tackle the problem, and a few algorithms (Patashnik et al., 2021; Tov et al., 2021) based on StyleGAN (Karras et al., 2019) exhibit outstanding performance. For instance, StyleCLIP (Patashnik et al., 2021) optimizes the latent vector of StyleGAN given by the source image by taking advantage of the pretrained Contrastive Language-Image Pretraining (CLIP) (Radford et al., 2021); it encourages the translated image using the updated latent vector to faithfully align well with the target domain.

Recently, with the rise of diffusion models (Sohl-Dickstein et al., 2015; Ho et al., 2020), existing methods for text-driven image editing employ the pretrained text-to-image diffusion models (Rombach et al., 2022; Ramesh et al., 2022; Saharia et al., 2022). To this end, diffusion-based approaches (Kim et al., 2022; Valevski et al., 2023; Kawar et al., 2023) perform an extra fine-tuning process on the pretrained diffusion models. Although they have shown to achieve promising results, these frameworks consume a significant amount of computation and memory due to the additional fine-tuning process. To the contrary, another line of research (Hertz et al., 2023; Tumanyan et al., 2023; Parmar et al., 2023) pursues training-free algorithms, which revise the original denoising process of the diffusion model without the additional fine-tuning process.

We propose a simple but effective training-free image-to-image translation method based on the pretrained diffusion models. Our method also modifies the original reverse process by optimizing two proposed guidance terms based on triplet loss with respect to the target latent variables. During target image generation, the objectives aim to preserve the structure of the source image while enhancing the fidelity to translation results. In addition to the guidance terms, we employ the cycle-consistency objective (Zhu et al., 2017) for the forward and backward processes to better preserve the structure or background of the source image; in this respect, our approach is distinct from the previous works focusing only on revising the denoising process, The main contributions of the proposed algorithm are summarized below:

- We propose simple but effective guidance terms using the representations given by the pretrained CLIP and Stable Diffusion models to enhance the backward process designed for text-driven image-to-image editing tasks.
- We employ the cycle-consistency objective for diffusion models to encourage the target images and preserve the structural and background information of the source images.
- Experimental results verify that our method achieves outstanding performance in various tasks compared with prior works.

The rest of this paper is organized as follows. Section 2 discusses the related prior works for text-to-image generation and diffusion-based image-to-image translation. We discuss the detailed framework of our proposed algorithm in Section 3. Experimental results are reported in Section 4, then we conclude our paper in Section 5.

## 2 RELATED WORK

### 2.1 TEXT-TO-IMAGE DIFFUSION MODELS

Existing methods (Rombach et al., 2022; Saharia et al., 2022; Ramesh et al., 2022) based on diffusion models have demonstrated remarkable performance on text-to-image generation tasks. For example, Stable Diffusion (Rombach et al., 2022) projects a given image into a low dimensional manifold using pre-trained auto-encoder frameworks (Kingma & Welling, 2014), and then estimates its distribution instead of the raw data distribution using the diffusion models. Imagen (Saharia et al., 2022) uses a large pre-trained text encoder to generate a text representation and synthesizes an image conditioned on the embedding based on the diffusion models. On the other hand, DALL·E 2 (Ramesh et al., 2022) introduces a prior network to estimate the CLIP image embedding from its corresponding text caption, and then generates an image conditioned on the two CLIP embeddings given the image and text caption pair.

### 2.2 TEXT-DRIVEN IMAGE MANIPULATION METHODS

Prior works attacking text-driven image manipulation tasks aim to preserve the structure and background of the source image while selectively editing the image to align with the target prompt. For example, using an unconditional diffusion model trained on ImageNet, DiffuseIT (Kwon & Ye, 2023) modifies the denoising process of Denoising Diffusion Probabilistic Models (DDPM) (Ho et al., 2020) based on the pre-trained networks of DINO ViT (Caron et al., 2021) and CLIP (Radford et al., 2021) . Different from DiffuseIT, several works (Parmar et al., 2023; Hertz et al., 2023; Tumanyan et al., 2023) employ the publicly available pre-trained Stable Diffusion to solve the text-driven image-to-image translation tasks. Specifically, Pix2pix-zero (Parmar et al., 2023) optimizes the target latent by making the cross-attention maps extracted from the pre-trained diffusion model given by source and target latents become similar. On the other hand, Prompt-to-Prompt (Hertz et al., 2023) injects the self-attention and cross-attention maps obtained from the source latents into those given by the target latents during the reverse process. Also, Plug-and-Play (Tumanyan et al., 2023) uses the self-attention and intermediate feature maps for the injection.

## 3 PROPOSED ALGORITHM

This section first describes the simple DDIM method based on diffusion models for text-driven image-to-image translation. Then, our sampling strategy that iteratively optimizes the forward and backward process is described in detail. Also, we present the proposed guidance objective based on a triplet loss that encourages a target image to align well with a target prompt while preserving the structure of the source image. Finally, we also discuss the cycle-consistency loss and its efficient version for the diffusion model enforced to improve the quality of translated results.

### 3.1 DDIM TARGET GENERATION

Diffusion models are proposed to estimate the data distribution. DDPM (Ho et al., 2020) assumes that the forward and backward processes follow a Markov chain. In the forward process, it gradu-

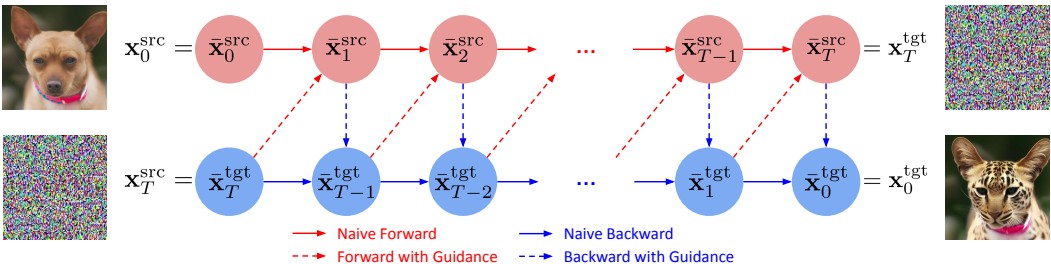

Figure 1: Overview of the proposed method about the forward guidance and backward guidance.

ally perturbs the original data with Gaussian noises while its intractable reverse transition kernel is approximated by modeling the Gaussian distribution whose mean is parametrized using the noise prediction network. On the other hand, Denoising Diffusion Implicit Models (DDIM) (Song et al., 2021) proposes the deterministic forward and backward processes, which makes the framework significantly reduce the number of denoising iterations without losing the generation quality.

For text-driven image-to-image translation tasks, existing algorithms often employ the deterministic process of DDIM using pre-trained text-to-image diffusion models. Specifically, they extract the final latent $\mathbf{x}_T^{\text{src}}$ recursively using the forward process starting from the source image $\mathbf{x}_0^{\text{src}}$, which is formally defined as

$$
\begin{aligned}
\mathbf{x}_{t+1}^{\text{src}} &= f_t^{\text{fwd}}(\mathbf{x}_t^{\text{src}}) \\
&= \sqrt{\alpha_{t+1}} \left( \frac{\mathbf{x}_t^{\text{src}} - \sqrt{1-\alpha_t}\epsilon_\theta(\mathbf{x}_t^{\text{src}}, t, \mathbf{y}^{\text{src}})}{\sqrt{\alpha_t}} \right) + \sqrt{1-\alpha_{t+1}}\epsilon_\theta(\mathbf{x}_t^{\text{src}}, t, \mathbf{y}^{\text{src}}),
\end{aligned} \quad (1)
$$

where $f_t^{\text{fwd}}(\cdot)$ denotes the naive forward process at timestep $t$ and $\epsilon_\theta(\cdot, \cdot, \cdot)$ is the noise prediction network while $\mathbf{x}_t^{\text{src}}$ and $\mathbf{y}^{\text{src}}$ are given source latent and source prompt corresponding to the source image $\mathbf{x}_0^{\text{src}}$.

Then, during the backward process, the target image $\mathbf{x}_0^{\text{tgt}}$ is synthesized from the final target latent $\mathbf{x}_T^{\text{tgt}}$ set equal to the final source latent $\mathbf{x}_T^{\text{src}}$ by recursively employing the following reverse DDIM process as

$$
\begin{aligned}
\mathbf{x}_{t-1}^{\text{tgt}} &= f_t^{\text{bwd}}(\mathbf{x}_t^{\text{tgt}}) \\
&= \sqrt{\alpha_{t-1}} \left( \frac{\mathbf{x}_t^{\text{tgt}} - \sqrt{1-\alpha_t}\epsilon_\theta(\mathbf{x}_t^{\text{tgt}}, t, \mathbf{y}^{\text{tgt}})}{\sqrt{\alpha_t}} \right) + \sqrt{1-\alpha_{t-1}}\epsilon_\theta(\mathbf{x}_t^{\text{tgt}}, t, \mathbf{y}^{\text{tgt}}),
\end{aligned} \quad (2)
$$

where $f_t^{\text{bwd}}(\cdot)$ is defined by the naive backward process at timestep $t$ while $\mathbf{x}^{\text{tgt}}$ and $\mathbf{y}^{\text{tgt}}$ are target latent and target prompt.

The naive DDIM translation, recursively using the backward process defined in Eq.2 from the final target latent $\mathbf{x}_T^{\text{tgt}}$, guarantees the cycle-consistency property as verified by Su et al. (2022). In other words, after the process of converting the source domain image $\mathbf{x}_0^{\text{src}}$ into the target domain $\mathbf{x}_0^{\text{tgt}}$ and then transforming it back to the source domain image denoted as $\hat{\mathbf{x}}_0^{\text{src}}$, the equality $\mathbf{x}_0^{\text{src}} = \hat{\mathbf{x}}_0^{\text{src}}$ holds. However, we empirically observe that the naive DDIM generation fails to preserve the overall structure and background of the source image in most cases. In order to tackle this problem, existing methods focus on revising the naive backward process only.

## 3.2 Proposed Target Generation

Different from previous frameworks (Kwon & Ye, 2023; Parmar et al., 2023; Hertz et al., 2023; Tumanyan et al., 2023), as shown in Figure 1, the proposed method alternately optimizes the source and target latents in the order of $\{\bar{\mathbf{x}}_{T-t}^{\text{src}}, \bar{\mathbf{x}}_t^{\text{tgt}}\}_{t=T-1:0}$, where $\bar{\mathbf{x}}_{T-t}^{\text{src}}$ and $\bar{\mathbf{x}}_t^{\text{tgt}}$ are source and target latents from our modified forward and backward processes. Note that the two latents are deviated from the naive DDIM samples $\mathbf{x}_{T-t}^{\text{src}}$ and $\mathbf{x}_t^{\text{tgt}}$. In the case of $t = T$, we note that $\bar{\mathbf{x}}_0^{\text{src}}$ is equal to $\mathbf{x}_0^{\text{src}}$

---

**Algorithm 1** Text-Driven Image Translation using Forward and Backward Guidances

---

**Inputs:** A source image $\mathbf{x}_0^{\text{src}}$, A source prompt $\mathbf{y}^{\text{src}}$, A target prompt $\mathbf{y}^{\text{tgt}}$, Hyperparamters $\lambda_1, \lambda_2, \lambda_3, \beta_1, \beta_2$

**for** $t \leftarrow 1, \cdots, T$ **do**

    Compute $\mathbf{x}_t^{\text{src}}$ using Eq. 1 while storing the feature map $F(\mathbf{x}_t^{\text{src}})$

**end for**

$\bar{\mathbf{x}}_T^{\text{tgt}} \leftarrow \mathbf{x}_T^{\text{src}}$ and $\bar{\mathbf{x}}_0^{\text{src}} \leftarrow \mathbf{x}_0^{\text{src}}$

**for** $t \leftarrow T, \cdots, 1$ **do**

    Compute $\mathcal{L}^{\text{cycle, eff}}$ using $\bar{\mathbf{x}}_t^{\text{tgt}}, \bar{\mathbf{x}}_{T-t}^{\text{src}}$ with Eq. 9-11 and 12

    Compute $\gamma'_{T-t} \leftarrow \sqrt{\frac{\alpha_{T-t+1}}{\alpha_{T-t}}} - \sqrt{\frac{1-\alpha_{T-t+1}}{1-\alpha_{T-t}}}$

    $\bar{\mathbf{x}}_{T-t+1}^{\text{src}} \leftarrow \sqrt{\frac{\alpha_{T-t+1}}{\alpha_{T-t}}}\bar{\mathbf{x}}_{T-t}^{\text{src}} - \sqrt{1-\alpha_{T-t}}\gamma'_{T-t}\epsilon_\theta(\bar{\mathbf{x}}_{T-t}^{\text{src}}, T-t, \mathbf{y}^{\text{src}}) - \nabla_{\bar{\mathbf{x}}_{T-t}^{\text{src}}}\lambda_1 \mathcal{L}^{\text{cycle, eff}}$

                                                 $\triangleright$ Forward with Guidance

    Compute $\mathcal{L}_t^{\text{dist}}$ using Eq. 6

    Compute $\mathcal{L}^{\text{cycle}}$ by calculating Eq. 7-11

    Compute $\gamma_t \leftarrow \sqrt{\frac{\alpha_{t-1}}{\alpha_t}} - \sqrt{\frac{1-\alpha_{t-1}}{1-\alpha_t}}$

    $\bar{\mathbf{x}}_{t-1}^{\text{tgt}} \leftarrow \sqrt{\frac{\alpha_{t-1}}{\alpha_t}}\bar{\mathbf{x}}_t^{\text{tgt}} - \sqrt{1-\alpha_t}\gamma_t\epsilon_\theta(\bar{\mathbf{x}}_t^{\text{tgt}}, t, \mathbf{y}^{\text{tgt}}) - \nabla_{\bar{\mathbf{x}}_t^{\text{tgt}}}(\lambda_2 \mathcal{L}^{\text{cycle}} + \lambda_3 \mathcal{L}_t^{\text{dist}})$

                                                 $\triangleright$ Backward with Guidance

**end for**

$\mathbf{x}_0^{\text{tgt}} \leftarrow \bar{\mathbf{x}}_0^{\text{tgt}}$

**Output:** A target image $\mathbf{x}_0^{\text{tgt}}$

---

which is the source image, while $\bar{\mathbf{x}}_T^{\text{tgt}}$ is equal to $\mathbf{x}_T^{\text{src}}$ that is obtained by recursively inverting the source image using Eq 1. Each modified process is denoted as *forward with guidance* and *backward with guidance*. The detailed procedures of the proposed guidances are summarized in Algorithm 1.

### 3.2.1 FORWARD WITH GUIDANCE

We revise the naive forward process in Eq. 1 by additionally optimizing the proposed efficient version of the cycle-consistency objective $\mathcal{L}^{\text{cycle, eff}}$ as described in Section 3.5 with respect to the source latent $\bar{\mathbf{x}}_{T-t}^{\text{src}}$ as follows:

$$
\begin{aligned}
\bar{\mathbf{x}}_{T-t+1}^{\text{src}} =& \bar{f}_{T-t}^{\text{fwd}}(\bar{\mathbf{x}}_{T-t}^{\text{src}}) \\
=& \sqrt{\frac{\alpha_{T-t+1}}{\alpha_{T-t}}}\bar{\mathbf{x}}_{T-t}^{\text{src}} - \sqrt{1-\alpha_{T-t}}\gamma'_{T-t}\epsilon_\theta(\bar{\mathbf{x}}_{T-t}^{\text{src}}, T-t, \mathbf{y}^{\text{src}}) - \nabla_{\bar{\mathbf{x}}_{T-t}^{\text{src}}}\lambda_1 \mathcal{L}^{\text{cycle, eff}},
\end{aligned} \quad (3)
$$

where $\bar{f}_{T-t}^{\text{fwd}}(\cdot)$ denotes the modified forward process at timestep $T-t$ while $\gamma'_{T-t}$ is denoted as $\sqrt{\frac{\alpha_{T-t+1}}{\alpha_{T-t}}} - \sqrt{\frac{1-\alpha_{T-t+1}}{1-\alpha_{T-t}}}$ and $\lambda_1$ is a hyperparameter.

### 3.2.2 BACKWARD WITH GUIDANCE

In backward with guidance, we enhance the naive backward process in Eq. 2 by optimizing the cycle-consistency objective $\mathcal{L}^{\text{cycle}}$ and the distance term $\mathcal{L}_t^{\text{dist}}$, where the modified sampling process is given as

$$
\begin{aligned}
\bar{\mathbf{x}}_{t-1}^{\text{tgt}} =& \bar{f}_t^{\text{bwd}}(\bar{\mathbf{x}}_t^{\text{tgt}}) \\
=& \sqrt{\frac{\alpha_{t-1}}{\alpha_t}}\bar{\mathbf{x}}_t^{\text{tgt}} - \sqrt{1-\alpha_t}\gamma_t\epsilon_\theta(\bar{\mathbf{x}}_t^{\text{tgt}}, t, \mathbf{y}^{\text{tgt}}) - \nabla_{\bar{\mathbf{x}}_t^{\text{tgt}}}(\lambda_2 \mathcal{L}^{\text{cycle}} + \lambda_3 \mathcal{L}_t^{\text{dist}}),
\end{aligned} \quad (4)
$$

where $\bar{f}_t^{\text{bwd}}(\cdot)$ is defined by our backward process at timestep $t$ while $\gamma_t$ is defined by $\sqrt{\frac{\alpha_{t-1}}{\alpha_t}} - \sqrt{\frac{1-\alpha_{t-1}}{1-\alpha_t}}$ and both $\lambda_2$ and $\lambda_3$ are hyperparameters. In the above equation, we will describe the two objectives $\mathcal{L}_t^{\text{dist}}$ and $\mathcal{L}^{\text{cycle}}$ in Section 3.3 and Section 3.4, respectively.

### 3.3 TRIPLET-BASED DISTANCE OBJECTIVE

For backward with guidance, we derive a naive distance term to guide the target latent $\bar{\mathbf{x}}_t^{\text{tgt}}$, where the objective is defined as

$$\mathcal{L}_t^{\text{naive-dist}} := -\text{Sim}(f^{\text{img}}(\bar{\mathbf{x}}_t^{\text{tgt}}), f^{\text{txt}}(\mathbf{y}^{\text{tgt}})) + \beta_1 \|F(\bar{\mathbf{x}}_t^{\text{tgt}}) - F(\mathbf{x}_t^{\text{src}})\|_{2,2}, \tag{5}$$

where $f^{\text{img}}(\cdot)$ and $f^{\text{txt}}(\cdot)$ are CLIP image and text encoders, respectively. In the above equation, $\text{Sim}(\cdot, \cdot)$ calculates the cosine similarity between two vectors, $\|\cdot\|_{2,2}$ is the $L_{2,2}$ matrix norm known as the Frobenius norm, $F(\cdot)$ is an intermediate feature map extracted from the noise prediction network of the T2I diffusion model, and $\beta_1$ is a hyperparameter. In case of using the second term of the right hand side in Eq. 5, it is motivated by the property that the feature maps extracted from the noise prediction network can capture the structural information of the given source image as demonstrated in Tumanyan et al. (2023).

In addition, motivated by the metric learning framework (Hoffer & Ailon, 2015), we enhance the naive objective in Eq. 5 by providing a harder constraint. Specifically, we translate the target image more closely aligned with the target prompt $\mathbf{y}^{\text{tgt}}$ compared with the source prompt $\mathbf{y}^{\text{src}}$. Also, since the reverse process destroys the structure of the source image, the distance of $F(\bar{\mathbf{x}}_t^{\text{tgt}})$ between $F(\mathbf{x}_t^{\text{src}})$ should be relatively closer compared to $F(\bar{\mathbf{x}}_{t+1}^{\text{tgt}})$ in order to the preserve the background or structure of the source image. Based on this motivation, we define an effective distance objective based on the triplet loss as

$$\begin{aligned}
\mathcal{L}_t^{\text{dist}} := \max(0, \; & \beta_1 \|F(\bar{\mathbf{x}}_t^{\text{tgt}}) - F(\mathbf{x}_t^{\text{src}})\|_{2,2} - \|F(\bar{\mathbf{x}}_t^{\text{tgt}}) - F(\bar{\mathbf{x}}_{t+1}^{\text{tgt}})\|_{2,2}) \\
& - \min(0, \; \text{Sim}(f^{\text{img}}(\bar{\mathbf{x}}_t^{\text{tgt}}), f^{\text{txt}}(\mathbf{y}^{\text{tgt}})) - \text{Sim}(f^{\text{img}}(\bar{\mathbf{x}}_t^{\text{tgt}}), f^{\text{txt}}(\mathbf{y}^{\text{src}})) - \beta_2),
\end{aligned} \tag{6}$$

where $\beta_2$ is also a hyperparameter.

### 3.4 CYCLE-CONSISTENCY OBJECTIVE

Although the naive DDIM translation guarantees the cycle-consistency property as we mentioned, the proposed method no longer guarantees the cycle-consistency property since we revised the generation process. We aim to recover it by employing the cycle-consistency objective to further enhance the performance. As described in CycleGAN (Zhu et al., 2017), the cycle-consistency term is formally defined as $\|\mathbf{x}_0^{\text{src}} - h(g(\mathbf{x}_0^{\text{src}}))\|_{2,2}$, where $g(\cdot)$ is the image-to-image translation operation from the source domain to the target domain, and vise versa for $h(\cdot)$. With the assumption that $g(\cdot)$ and $h(\cdot)$ are invertible, we alternatively optimize the following cycle-consistency objective, which is given by

$$\mathcal{L}^{\text{cycle}} := \|\bar{\mathbf{x}}_{0,f}^{\text{tgt}} - \bar{\mathbf{x}}_{0,b}^{\text{tgt}}\|_{2,2}, \tag{7}$$

where we denote $\bar{\mathbf{x}}_{0,f}^{\text{tgt}}$ as $h^{-1}(\mathbf{x}_0^{\text{src}})$ and $\bar{\mathbf{x}}_{0,b}^{\text{tgt}}$ as $g(\mathbf{x}_0^{\text{src}})$. The definition of $h^{-1}(\cdot)$ and $g(\cdot)$ are as

$$\bar{\mathbf{x}}_{0,f}^{\text{tgt}} = h^{-1}(\mathbf{x}_0^{\text{src}}) = F^{\text{bwd}}(\bar{F}^{\text{fwd}}(\mathbf{x}_0^{\text{src}})), \quad \bar{\mathbf{x}}_{0,b}^{\text{tgt}} = g(\mathbf{x}_0^{\text{src}}) = \bar{F}^{\text{bwd}}(F^{\text{fwd}}(\mathbf{x}_0^{\text{src}})), \tag{8}$$

where $F^{\text{fwd}}(\cdot)$ and $F^{\text{bwd}}(\cdot)$ are denoted by $f_{T-1}^{\text{fwd}} \circ f_{T-2}^{\text{fwd}} \cdots \circ f_0^{\text{fwd}}(\cdot)$ and $f_1^{\text{bwd}} \circ f_2^{\text{bwd}} \cdots \circ f_T^{\text{bwd}}(\cdot)$ while $\bar{F}^{\text{fwd}}(\cdot)$ and $\bar{F}^{\text{bwd}}(\cdot)$ are defined by $\bar{f}_{T-1}^{\text{fwd}} \circ \bar{f}_{T-2}^{\text{fwd}} \cdots \circ \bar{f}_0^{\text{fwd}}(\cdot)$ and $\bar{f}_1^{\text{bwd}} \circ \bar{f}_2^{\text{bwd}} \cdots \circ \bar{f}_T^{\text{bwd}}(\cdot)$.

**Estimation of $\bar{\mathbf{x}}_{0,f}^{\text{tgt}}$** In order to reduce the computation of $\bar{\mathbf{x}}_{0,f}^{\text{tgt}}$, using the Tweedie's formula from Stein (1981), we infer the denoised image $\bar{\mathbf{x}}_0^{\text{src}}$ from $\bar{\mathbf{x}}_{T-t}^{\text{src}}$ as follows:

$$\bar{\mathbf{x}}_0^{\text{src}} = \frac{\bar{\mathbf{x}}_{T-t}^{\text{src}} - \sqrt{1 - \alpha_{T-t}} \, \epsilon_\theta(\bar{\mathbf{x}}_{T-t}^{\text{src}}, T-t, \mathbf{y}^{\text{src}})}{\sqrt{\alpha_{T-t}}}. \tag{9}$$

Using the equivalent ordinary differential equation of the naive DDIM forward process in Eq. 1, we approximate $\bar{\mathbf{x}}_{T,f}^{\text{tgt}}$, which is equal to $\bar{\mathbf{x}}_T^{\text{src}}$ that can be obtained by substituting Eq. 9 into the first term in the right hand side of Eq. 1 (*i.e.* the prediction of $\bar{\mathbf{x}}_0^{\text{src}}$ given $\bar{\mathbf{x}}_{T-t}^{\text{src}}$) as follows:

$$\bar{\mathbf{x}}_{T,f}^{\text{tgt}} = \bar{\mathbf{x}}_T^{\text{src}} = \sqrt{\frac{\alpha_T}{\alpha_{T-t}}} \bar{\mathbf{x}}_{T-t}^{\text{src}} + \left( \sqrt{1 - \alpha_T} - \sqrt{\frac{\alpha_T (1 - \alpha_{T-t})}{\alpha_{T-t}}} \right) \epsilon_\theta(\bar{\mathbf{x}}_{T-t}^{\text{src}}, T-t, \mathbf{y}^{\text{src}}). \tag{10}$$

In the above equation, we use the Euler method to solve the equivalent ordinary differential equation. Although it incurs discretization errors due to the one-step inversion from $T-t$ to $T$, we empirically observe that the proposed method achieves remarkable performance as demonstrated in Section 4. Finally, $\bar{\mathbf{x}}_{0,f}^{\text{tgt}}$ can be obtained by $F^{\text{bwd}}(\bar{\mathbf{x}}_{T,f}^{\text{tgt}})$, where $F^{\text{bwd}}(\cdot)$ performs the $T$ steps of recursive reverse DDIM sampling in Eq. 2.

**Estimation of $\bar{\mathbf{x}}_{0,b}^{\text{tgt}}$**   In order to reduce the computation of $\bar{\mathbf{x}}_{0,b}^{\text{tgt}}$, we approximate $\bar{\mathbf{x}}_{0,b}^{\text{tgt}}$ using the Tweedie's formula (Stein, 1981), which is given by

$$\bar{\mathbf{x}}_{0,b}^{\text{tgt}} = \frac{\bar{\mathbf{x}}_t^{\text{tgt}} - \sqrt{1-\alpha_t}\epsilon_\theta(\bar{\mathbf{x}}_t^{\text{tgt}}, t, \mathbf{y}^{\text{tgt}})}{\sqrt{\alpha_t}}. \tag{11}$$

We eventually can calculate $\mathcal{L}^{\text{cycle}}$ by plugging $\bar{\mathbf{x}}_{0,f}^{\text{tgt}}$ and $\bar{\mathbf{x}}_{0,b}^{\text{tgt}}$ into Eq. 7.

## 3.5   Efficient Cycle-Consistency Objective

In case of forward with guidance, the gradient computation of $\mathcal{L}_{\text{cycle}}$ using Eq. 7 with respect to the source latent $\bar{\mathbf{x}}_{T-t}^{\text{src}}$ is expensive in terms of memory and time since it involves multiple times of backpropagation through the noise prediction network. To tackle this issue, we alternatively derive the following efficient version of the cycle-consistency objective that matches the final target latents instead of the target images as

$$\mathcal{L}^{\text{cycle, eff}} := \|\bar{\mathbf{x}}_{T,f}^{\text{tgt}} - \bar{\mathbf{x}}_{T,b}^{\text{tgt}}\|_{2,2}. \tag{12}$$

**Estimation of $\bar{\mathbf{x}}_{T,f}^{\text{tgt}}$**   In the above equation, $\bar{\mathbf{x}}_{T,f}^{\text{tgt}}$ is obtained from Eq. 10 using a single forward propagation of the noise prediction network.

**Estimation of $\bar{\mathbf{x}}_{T,b}^{\text{tgt}}$**   $\bar{\mathbf{x}}_{T,b}^{\text{tgt}}$ is calculated by $F^{\text{fwd}}(\bar{\mathbf{x}}_{0,b}^{\text{tgt}})$, where $\bar{\mathbf{x}}_{0,b}^{\text{tgt}}$ is obtained by using Eq. 11, and $F^{\text{fwd}}(\cdot)$ applies the naive DDIM forward process in Eq. 1 for $T$ times.

Therefore, we can compute the gradient of $\mathcal{L}^{\text{cycle, eff}}$ with respect to $\bar{\mathbf{x}}_{T-t}^{\text{src}}$ just by performing a single backpropagation through the noise prediction network.

## 4   Experiments

We compare the proposed algorithm with state-of-the-art methods (Parmar et al., 2023; Hertz et al., 2023; Tumanyan et al., 2023) using the pretrained Stable Diffusion (Rombach et al., 2022). Also, we present the ablation study to quantitatively and qualitatively analyze the effects of each component.

### 4.1   Implementation Details

We implement the proposed method using the publicly available code of Pix2Pix-Zero (Parmar et al., 2023) based on PyTorch. In order to speed up the translation of the given source images, we reduce the number of denoising timesteps to 50 for all comparison algorithms including the proposed method. Also, we replace the original captions with generated captions given by Bootstrapping Language-Image Pre-training (BLIP) (Li et al., 2022), as the original captions often include languages other than English while the target prompts are constructed from the source prompts according to tasks. As an example, in case of the cat-to-dog task, we replace the word in the source prompt most closely related to 'cat' based on the CLIP text encoder with 'dog'. Note that we execute the official codes of Prompt-to-Prompt[1] (Hertz et al., 2023), Plug-and-Play[2] (Tumanyan et al., 2023), and Pix2Pix-Zero[3] (Parmar et al., 2023) using the same source and target prompts combined with the classifier-free guidance (Ho & Salimans, 2021).

---

[1]https://github.com/google/prompt-to-prompt
[2]https://github.com/MichalGeyer/plug-and-play
[3]https://github.com/pix2pixzero/pix2pix-zero

Table 1: Quantitative results to compare the proposed method with Prompt-to-Prompt (Hertz et al., 2023), Plug-and-Play (Tumanyan et al., 2023), and Pix2Pix-Zero (Parmar et al., 2023) using the pre-trained Stable Diffusion (Rombach et al., 2022) and real images sampled from the LAION-5B dataset (Schuhmann et al., 2022) for various tasks. For the the drawing → oil painting task, we do not report the BD score since the background can not be clearly defined. Black and red bold-faced numbers represent the best and second-best performance in each row.

| Task | Prompt-to-Prompt | | | Plug-and-Play | | | Pix2Pix-Zero | | | Ours | | |
|---|---|---|---|---|---|---|---|---|---|---|---|---|
| | CS (↑) | SD (↓) | BD (↓) | CS (↑) | SD (↓) | BD (↓) | CS (↑) | SD (↓) | BD (↓) | CS (↑) | SD (↓) | BD (↓) |
| cat → dog | 0.295 | 0.034 | 0.203 | 0.273 | **0.023** | 0.224 | **0.300** | 0.030 | **0.189** | **0.299** | 0.019 | 0.137 |
| dog → cat | 0.293 | 0.031 | **0.117** | 0.278 | **0.019** | 0.121 | **0.295** | 0.034 | 0.144 | **0.296** | 0.018 | 0.084 |
| dog → crochet dog | **0.305** | 0.024 | **0.098** | 0.293 | **0.021** | 0.122 | 0.303 | 0.029 | 0.123 | **0.315** | 0.011 | 0.070 |
| horse → zebra | **0.316** | 0.033 | 0.095 | 0.309 | **0.022** | **0.067** | **0.316** | 0.047 | 0.160 | **0.325** | 0.021 | 0.057 |
| drawing → oil painting | 0.287 | 0.043 | - | **0.288** | **0.023** | - | 0.259 | 0.035 | - | **0.293** | 0.014 | - |

## 4.2 EXPERIMENTAL SETTINGS

**Dataset and Tasks** We select about 250 images for all tasks from the LAION-5B dataset (Schuhmann et al., 2022) based on the CLIP similarity with their captions (Beaumont, 2022) in order to evaluate text-driven image editing performance. We chose five tasks for evaluating text-driven image-to-image translation performance. For the tasks, we consider object-centric tasks such as translating cats, dogs, and horses into dogs, cats, and zebra denoted as cat → dog, dog → cat, horse→ zebra, respectively. Also, we choose the dog → crochet dog task, where we selectively editing to make the dog look like the one made from yarn. Additionally, we test the proposed approach by transforming hand-drawn sketches into oil paintings denoted as drawing → oil painting.

**Evaluation Metrics** We both analyze 1) how well the synthesized target image aligns with the target prompt, and 2) how well the structure of the source image is preserved while translation. First of all, we measure the similarity between target prompt and generated target image using CLIP (Radford et al., 2021), which we call CLIP Similarity (**CS**). Secondly, to measure the overall structure difference between source and target image, we utilize the self-similarity map of ViT (Tumanyan et al., 2022) extracted from the source and target image. Using the feature maps, we calculate the squared Euclidean distance between them denoted by Structure Distance (**SD**). In addition, to measure how well each algorithm preserves the background, we delete the object parts of each source and target image using pre-trained segmentation model Detic (Zhou et al., 2022). Then, we measure the squared Euclidean distance between the background parts of the two images, which is referred to as Background Distance (**BD**).

## 4.3 QUANTITATIVE RESULTS

We report the quantitative results in Table 1 to compare with state-of-the-art methods (Parmar et al., 2023; Hertz et al., 2023; Tumanyan et al., 2023) on various tasks using the pre-trained Stable Diffusion and images sampled from the LAION-5B dataset. As presented in the table, our method outperforms the comparison algorithms for all metrics including CS, SD, and BD except the only one case, which implies that the proposed method is effective for the text-driven image-to-image translation tasks. In case of the drawing → oil painting task, we do not report the BD score since the background can not be clearly defined.

## 4.4 QUALITATIVE RESULTS

We visualize translated results in Figure 2 given by Prompt-to-Prompt (Hertz et al., 2023), Plug-and-Play (Tumanyan et al., 2023), Pix2Pix-Zero (Parmar et al., 2023), and the proposed method using the pre-trained Stable Diffusion and real images sampled from the LAION-5B dataset on various tasks. As presented in the figure, the proposed method achieves remarkable text-driven image-to-image translation performance compared with other methods. Different from our method, other algorithms do not preserve the structure well considering the source and translated images.

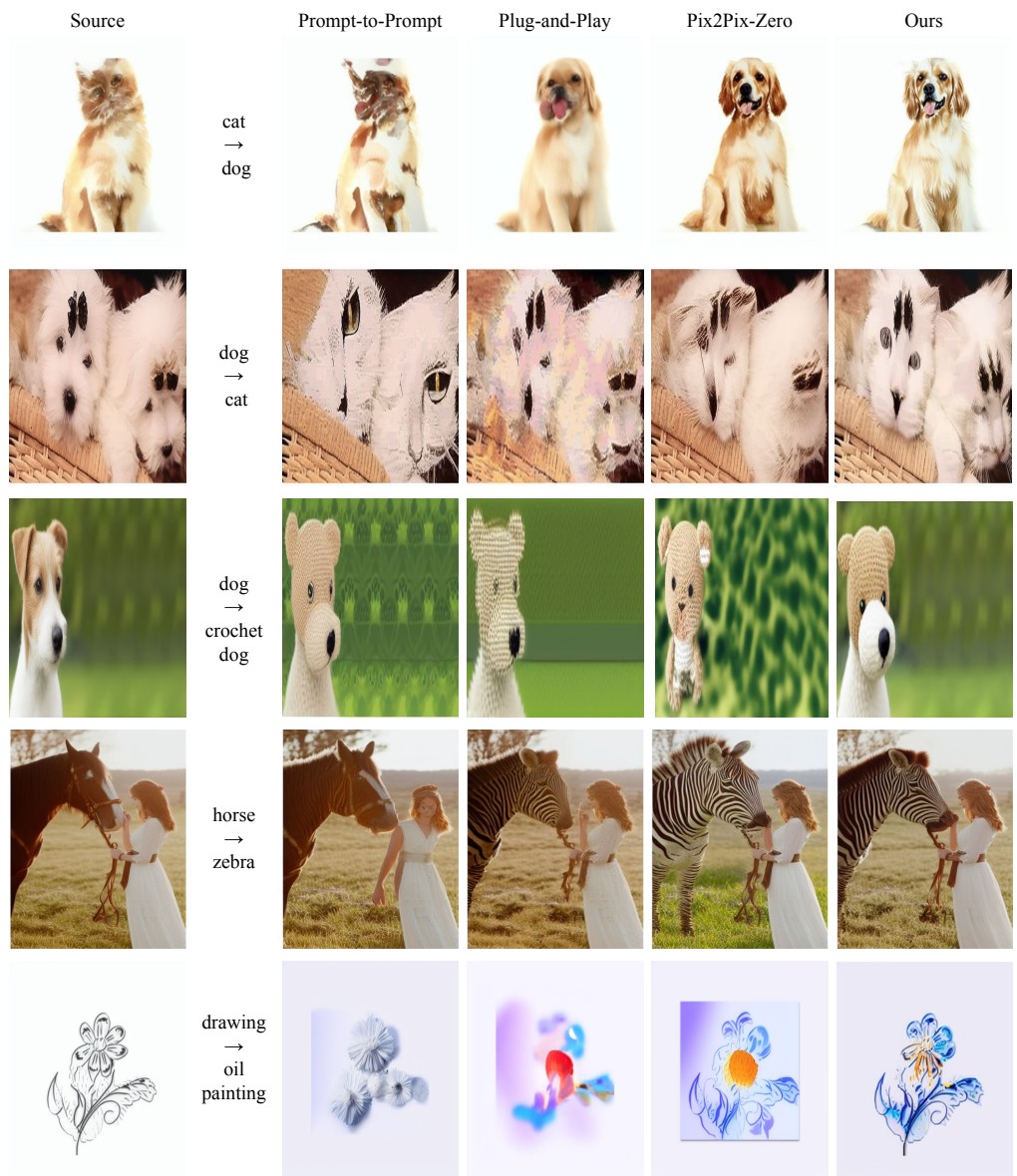

Figure 2: Qualitative results of the proposed method and other text-driven image-to-image translation methods (Hertz et al., 2023; Tumanyan et al., 2023; Parmar et al., 2023) using the pretrained Stable Diffusion (Rombach et al., 2022) and real images sampled from the LAION-5B dataset (Schuhmann et al., 2022) on various tasks.

## 4.5 ABLATION STUDY

In order to analyze the effect of each proposed component, we perform an ablation study on various tasks using the Stable Diffusion and images sampled from the LAION-5B dataset. As illustrated in Table 2, Naive Distance defined in Eq. 5 is helpful to increase text-driven image editing performance compared with DDIM. Additionally, Distance objective based on triplet loss defined in Eq. 6 leads to reduce the background distance while the cycle-consistency term can further minimize the background and structure distance. Furthermore, we visualize the qualitative results in Figure 3, which implies that each component is helpful for text-driven image-to-image translation tasks.

Table 2: Ablation study results to analyze the effect of each proposed component using the pre-trained Stable Diffusion (Rombach et al., 2022) and real images sampled from the LAION-5B dataset (Schuhmann et al., 2022) for various tasks. DDIM employs the reverse process defined in Eq. 2 while Naive Distance replaces the triplet distance term in Eq. 6 with the naive distance term in Eq. 5. Note that DDIM, Naive Distance, and Distance do not employ our cycle consistency objective.

| Task | DDIM | | | Naive Distance | | | Distance | | | Distance + Cycle (Ours) | | |
|---|---|---|---|---|---|---|---|---|---|---|---|---|
| | CS (↑) | SD (↓) | BD (↓) | CS (↑) | SD (↓) | BD (↓) | CS (↑) | SD (↓) | BD (↓) | CS (↑) | SD (↓) | BD (↓) |
| cat → dog | 0.289 | 0.072 | 0.347 | **0.298** | **0.020** | 0.159 | **0.298** | **0.020** | **0.153** | **0.299** | **0.019** | **0.137** |
| dog → cat | 0.289 | 0.063 | 0.224 | **0.297** | 0.020 | **0.087** | **0.297** | **0.019** | 0.084 | **0.296** | **0.018** | **0.084** |
| dog → crochet dog | 0.299 | 0.069 | 0.242 | **0.316** | 0.017 | 0.080 | **0.315** | **0.015** | **0.071** | **0.315** | **0.011** | **0.070** |
| horse → zebra | 0.310 | 0.081 | 0.255 | 0.322 | **0.023** | 0.081 | **0.324** | 0.026 | **0.067** | **0.325** | **0.021** | **0.057** |
| drawing → oil painting | 0.269 | 0.128 | - | **0.292** | 0.016 | - | **0.293** | **0.013** | - | **0.293** | **0.014** | - |

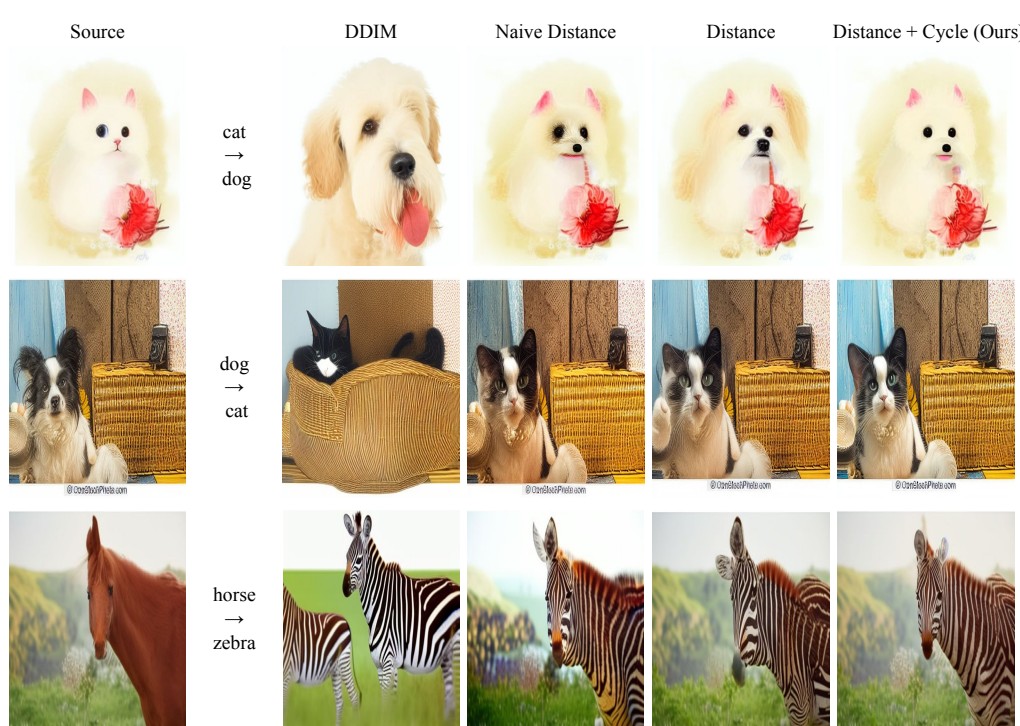

Figure 3: Qualitative results of our components using the pre-trained Stable Diffusion (Rombach et al., 2022) and data from the LAION-5B dataset (Schuhmann et al., 2022) on various tasks.

## 5 CONCLUSION

We proposed a simple but effective method for text-driven image-to-image translation tasks based on pre-trained text-to-image diffusion models. Although the naive DDIM method meets the cycle-consistency property, we empirically observe that it destroys significantly the structure of the source image during target image generation. In order to address the issue, we propose the distance objective based on triplet loss to revise the denoising process. For the derivation of the distance objective, we encourage the target latents to align well with the target prompts compared with the source prompts for fidelity while the target latents become closer to the source latents rather than the previous target latents for preserving the structure or background of the source image. Due to the additional revised term, our generation process no longer meets the cycle-consistency property. In order to further enhance the generation quality, we aim to recover the property. Experimental results demonstrate that our framework achieves outstanding performance qualitatively and quantitatively on various tasks when combined with the pre-trained Stable Diffusion.

**Code of Ethics**   The proposed method can generate harmful or misleading samples due to the pre-trained model. For example, the pre-trained network can generate realistic samples that can potentially violate the privacy.

**Reproducibility**   The source and target prompts are obtained using the generated captions given by BLIP (Li et al., 2022), where the detailed procedures are discussed in Section 4.1, For CS, SD, and BD as the quantitative evaluation metrics, we employ the pre-trained CLIP (Radford et al., 2021), ViT (Tumanyan et al., 2022), and Detic (Zhou et al., 2022), respectively as we mentioned in Section 4.2.

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

# A APPENDIX

## A.1 ADDITIONAL QUALITATIVE RESULTS

In addition to Figure 2 in the main paper, we present additional qualitative results in Figure 4, 5, 6, 7, and 8 to compare with state-of-the-art methods (Hertz et al., 2023; Tumanyan et al., 2023; Parmar et al., 2023) for text-driven image manipulation tasks using the pre-trained Stable Diffusion (Rombach et al., 2022) and real images sampled from the LAION-5B dataset (Schuhmann et al., 2022). As presented in the figures, the proposed method achieves qualitative better results than previous methods. In order to demonstrate the generalizability of the proposed method, we perform additional experiments using the cat and dog categories of Animal FacesHQ (AFHQ) datasets (Choi et al., 2020) and a synthetic dataset given by the Stable Diffusion on various tasks. As visualized in Figure 9, 10, and 11, our method also achieves outstanding text-driven image manipulation performance on the AFHQ and synthetic datasets.

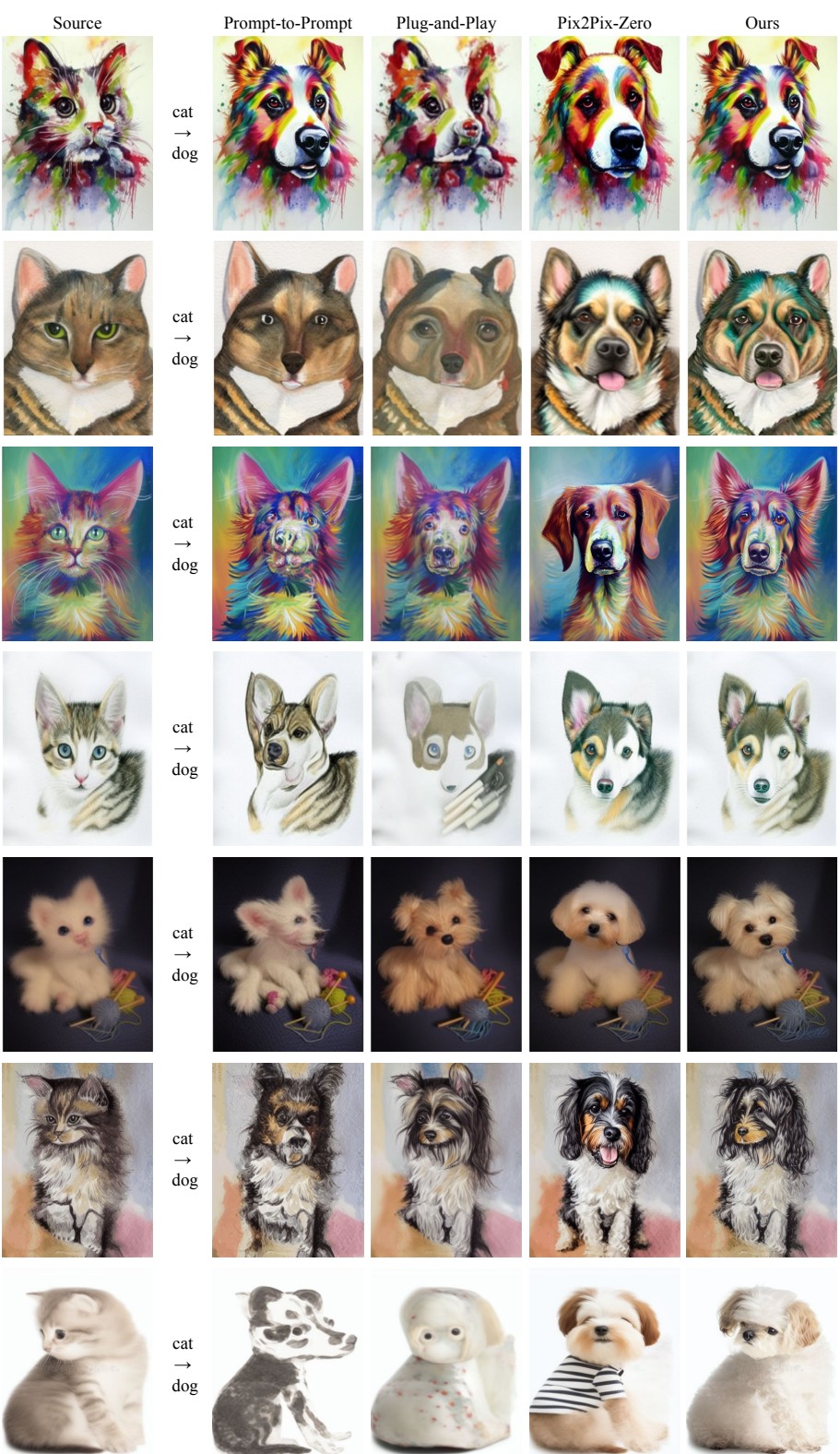

Figure 4: Additional qualitative results of the proposed method and other text-driven image-to-image translation methods (Hertz et al., 2023; Tumanyan et al., 2023; Parmar et al., 2023) using the pre-trained Stable Diffusion (Rombach et al., 2022) and real images sampled from the LAION 5B dataset (Schuhmann et al., 2022) on the cat → dog task.

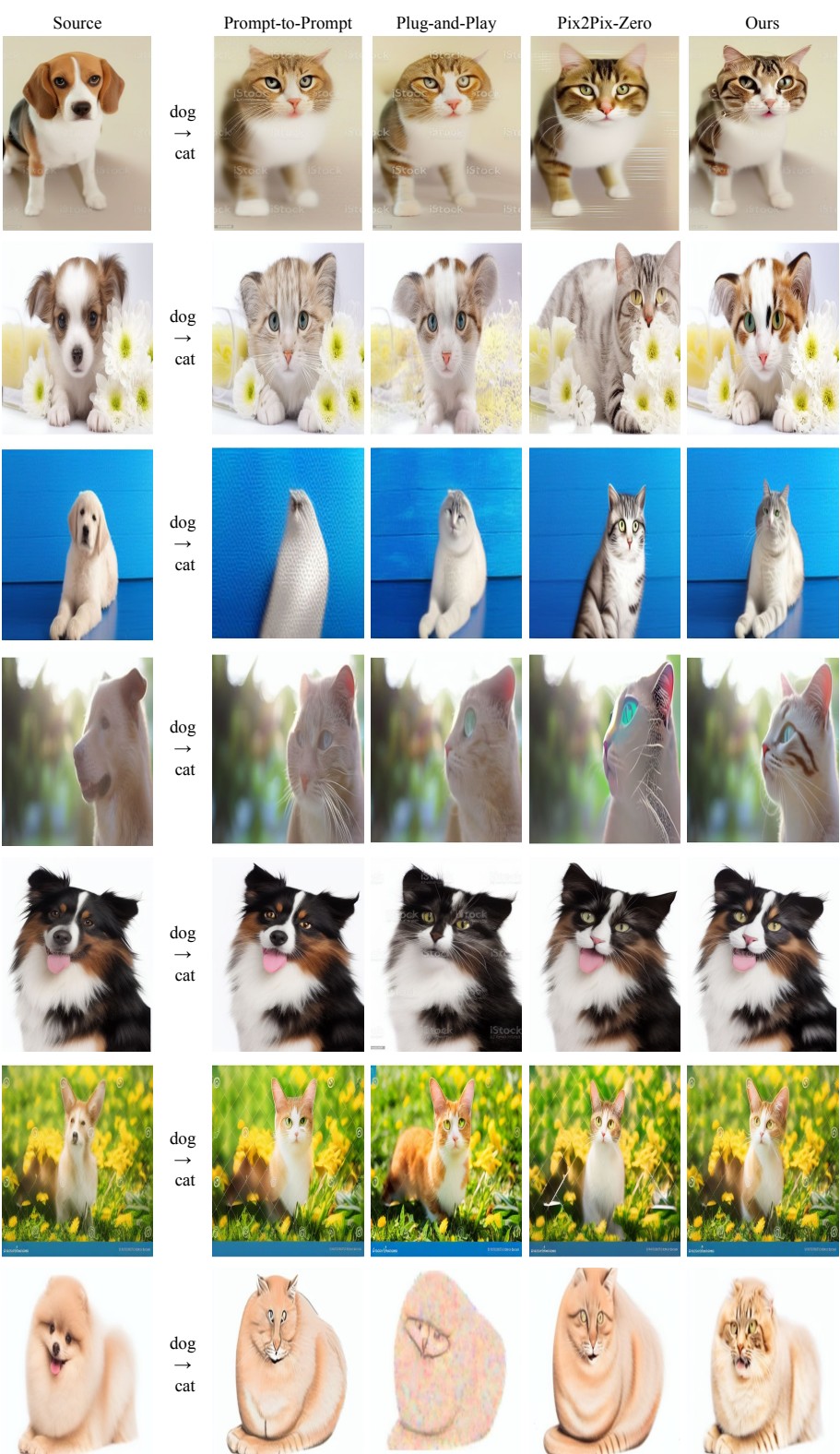

Figure 5: Additional qualitative results of the proposed method and other text-driven image-to-image translation methods (Hertz et al., 2023; Tumanyan et al., 2023; Parmar et al., 2023) using the pre-trained Stable Diffusion (Rombach et al., 2022) and real images sampled from the LAION 5B dataset (Schuhmann et al., 2022) on the dog → cat task.

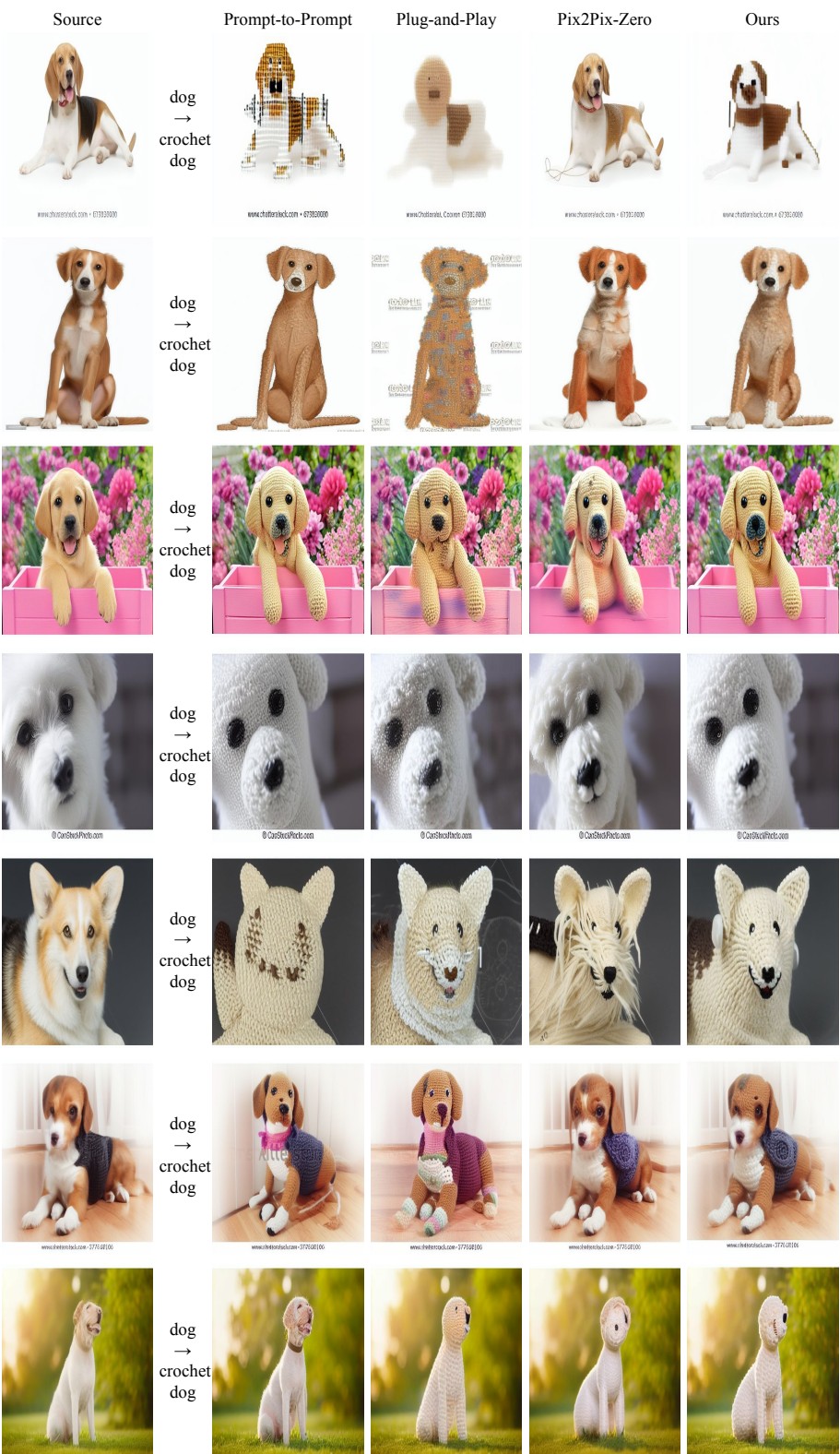

Figure 6: Additional qualitative results of the proposed method and other text-driven image-to-image translation methods (Hertz et al., 2023; Tumanyan et al., 2023; Parmar et al., 2023) using the pre-trained Stable Diffusion (Rombach et al., 2022) and real images sampled from the LAION 5B dataset (Schuhmann et al., 2022) on the dog → crochet dog task.

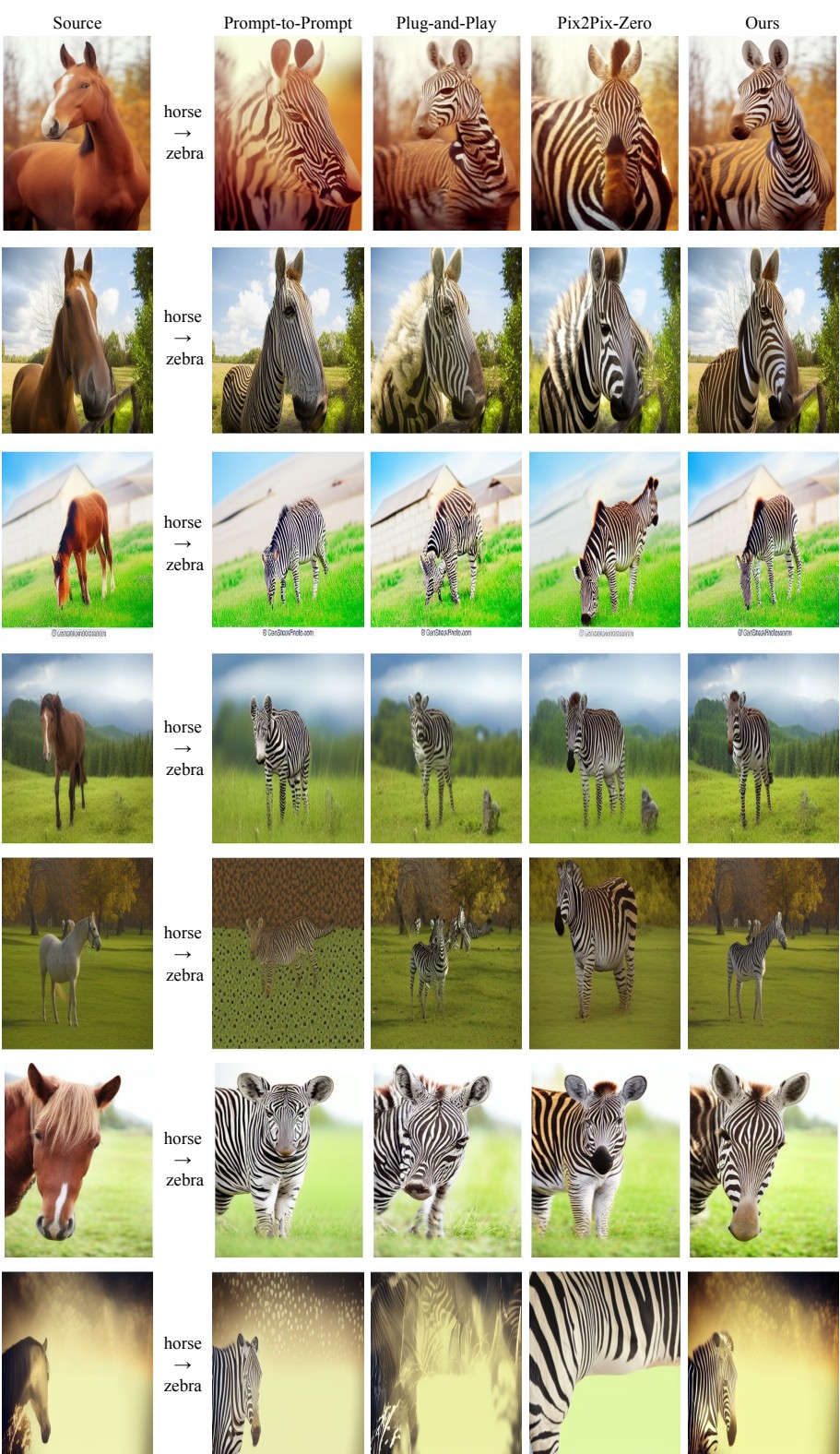

Figure 7: Additional qualitative results of the proposed method and other text-driven image-to-image translation methods (Hertz et al., 2023; Tumanyan et al., 2023; Parmar et al., 2023) using the pre-trained Stable Diffusion (Rombach et al., 2022) and real images sampled from the LAION 5B dataset (Schuhmann et al., 2022) on the horse → zebra task.

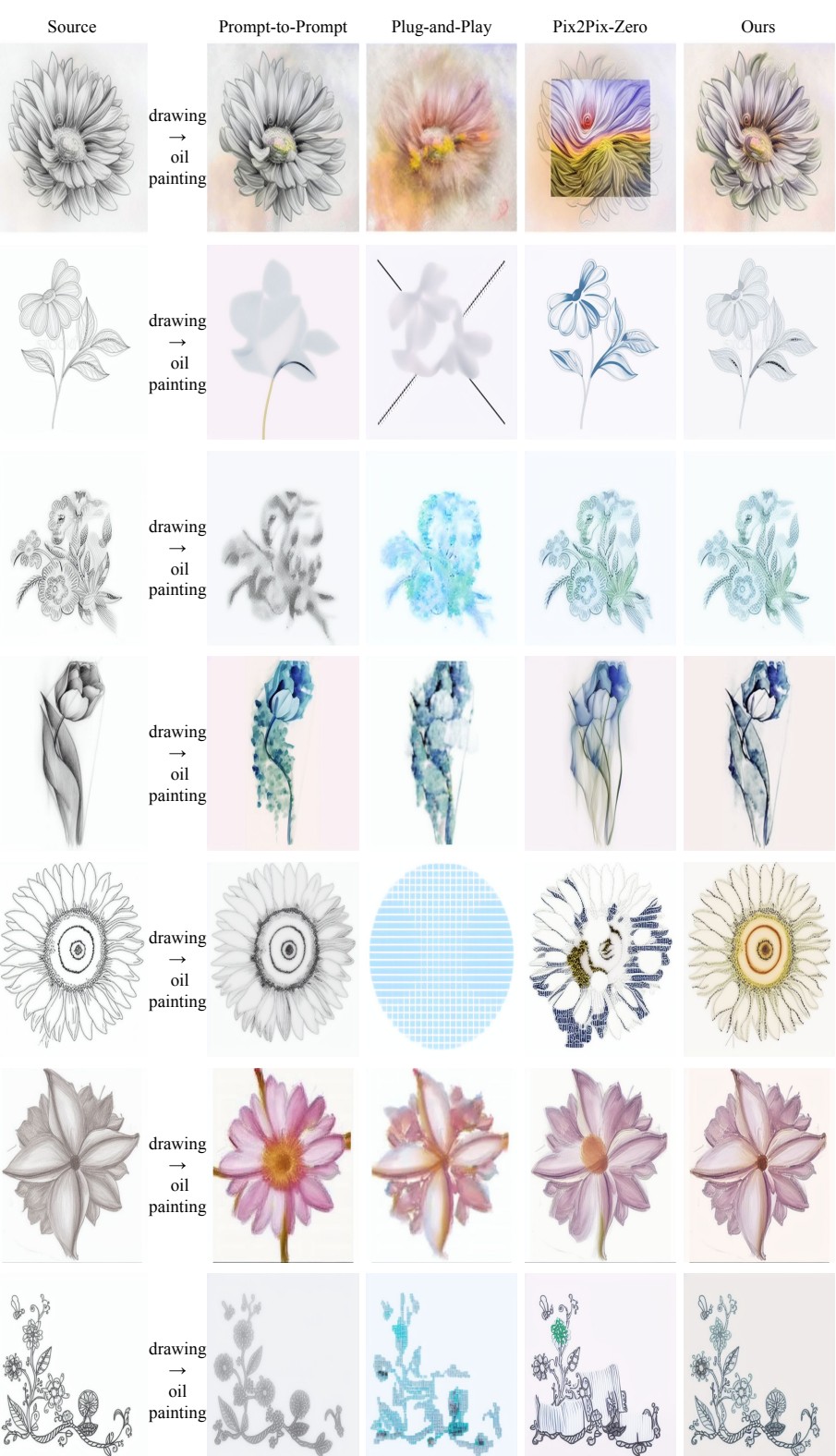

Figure 8: Additional qualitative results of the proposed method and other text-driven image-to-image translation methods (Hertz et al., 2023; Tumanyan et al., 2023; Parmar et al., 2023) using the pre-trained Stable Diffusion (Rombach et al., 2022) and real images sampled from the LAION 5B dataset (Schuhmann et al., 2022) on the drawing → oil painting task.

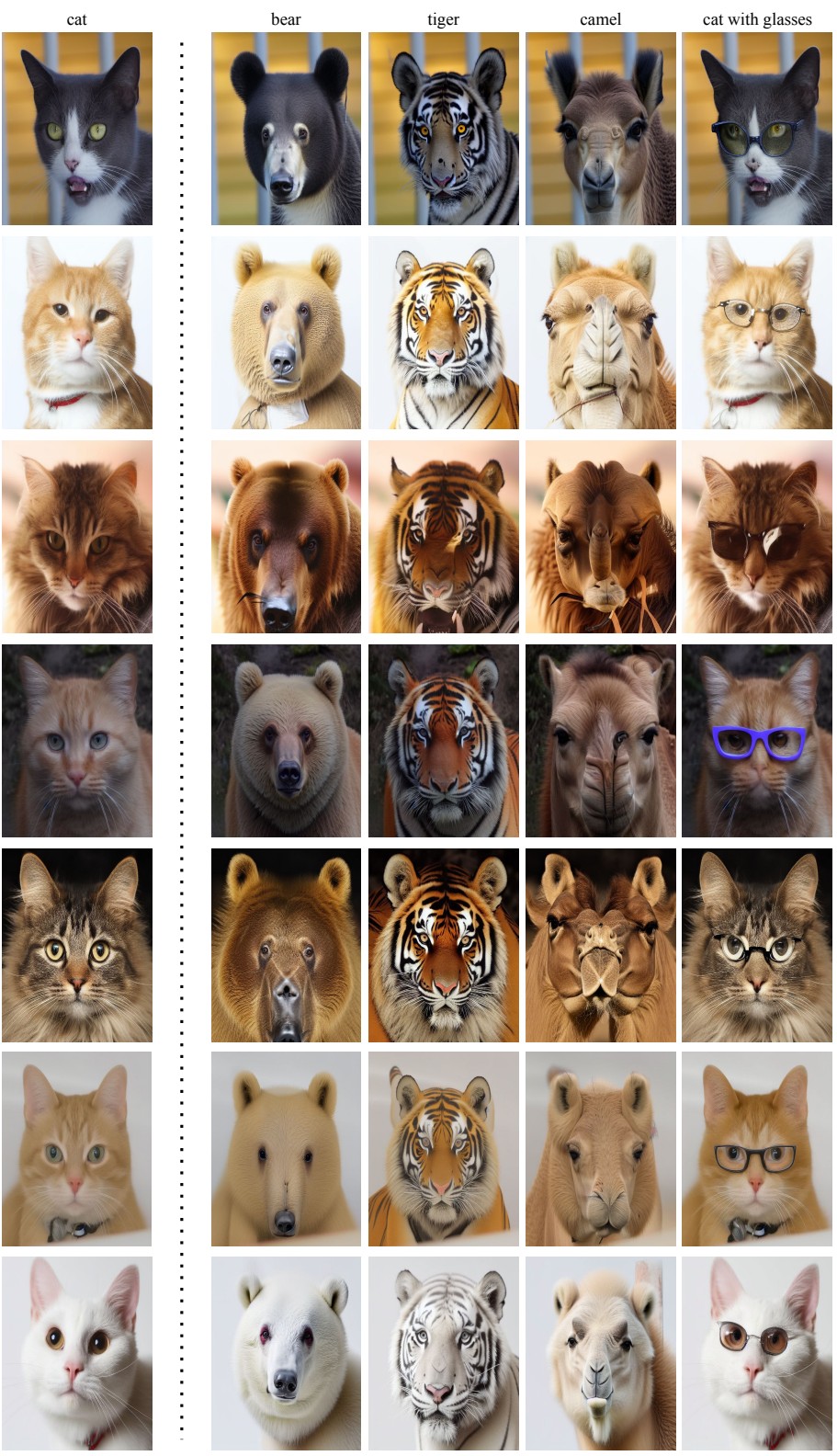

Figure 9: Qualitative results of the proposed method using the pre-trained Stable Diffusion (Rombach et al., 2022) and real images sampled from the cat of AFHQ dataset (Choi et al., 2020) on various tasks.

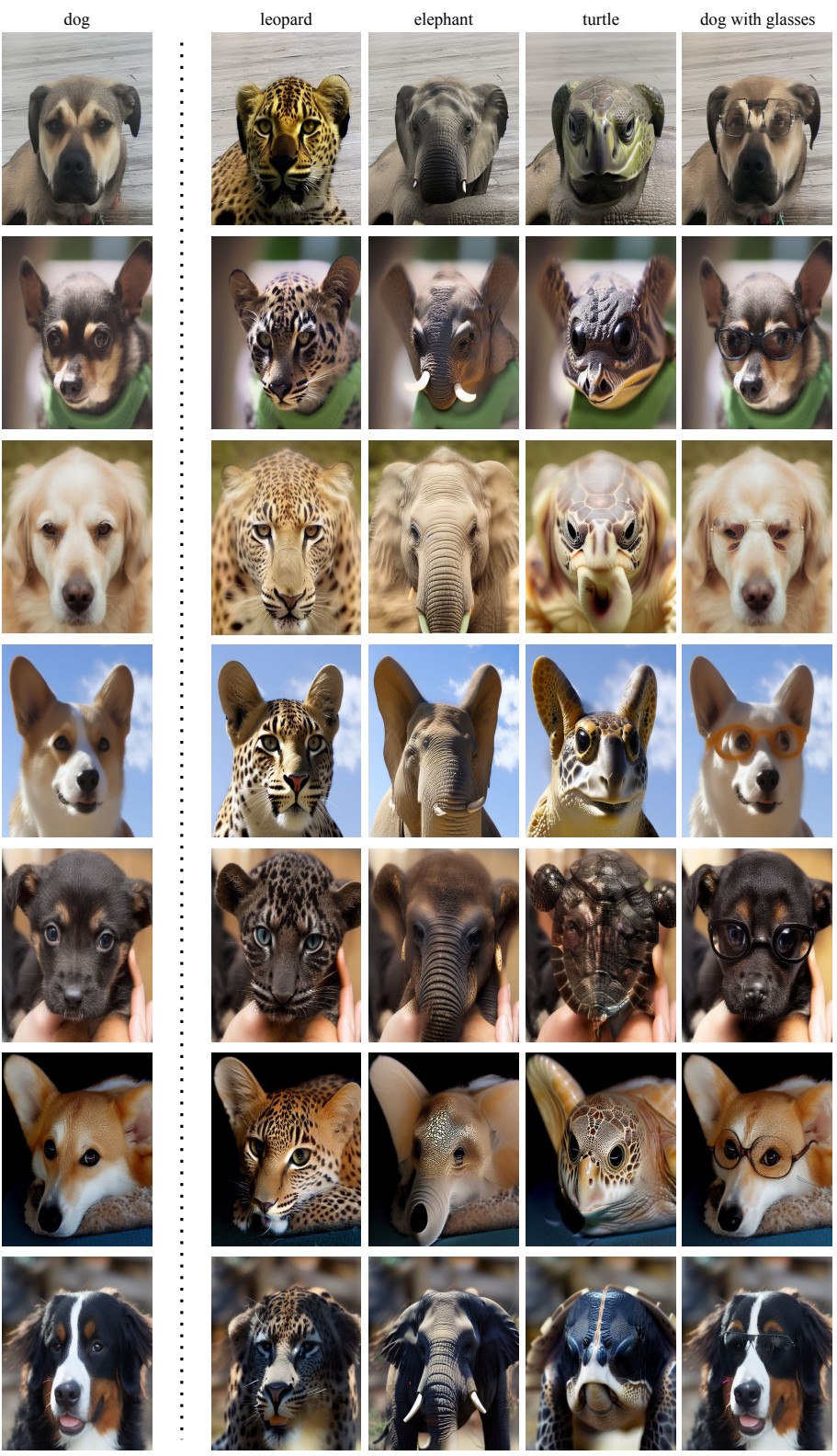

Figure 10: Qualitative results of the proposed method using the pre-trained Stable Diffusion (Rombach et al., 2022) and real images sampled from the dog of AFHQ dataset (Choi et al., 2020) on various tasks.

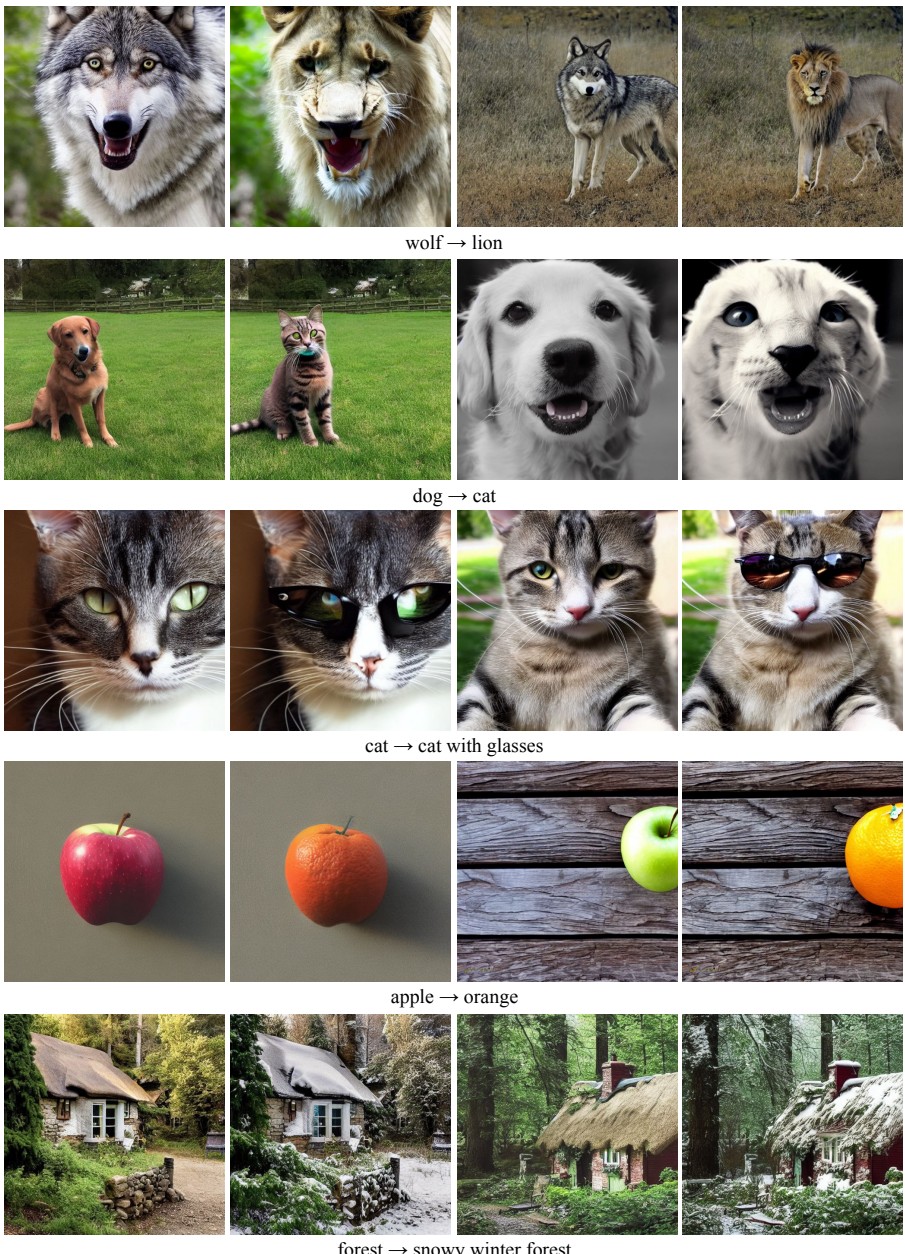

wolf → lion

dog → cat

cat → cat with glasses

apple → orange

forest → snowy winter forest

Figure 11: Qualitative results of the proposed method using the pre-trained Stable Diffusion (Rombach et al., 2022) and its synthesized images on various tasks.

