# OpenReview forum: "Text-Driven Image Editing using Cycle-Consistency-Driven Metric Learning"
_ICLR.cc/2024/Conference — ICLR 2024 Conference Withdrawn Submission_

### Official Review · Reviewer_k61j · 2023-10-29

**Soundness:** 2 fair
**Presentation:** 2 fair
**Contribution:** 2 fair
**Rating:** 5
**Confidence:** 4

**Summary:**

This work presents a training-free approach for text-driven image-to-image translation, building on a pre-trained text-to-image diffusion model. The authors revise the process to align better with the target task. They introduce a new guidance objective, which combines maximizing similarity to the target prompt (measured by CLIP score) and minimizing the distance to the source latent variables. Moreover, they employ a cycle-consistency objective to maintain the source image background by iteratively optimizing source and target latent variables. Experimental results demonstrate the exceptional performance of this method.

**Strengths:**

The article introduces a simple yet effective approach for text-driven image-to-image translation.
1. In contrast to other methods, this approach places a strong emphasis on preserving the structure and background of the source image during image editing. It accomplishes this by revising the process for generating target images to better align with the target task.
2.  The article introduces a new guidance objective that combines maximizing similarity to the target prompt (measured by CLIP scores) and minimizing the distance to the source latent variables, resulting in improved quality of generated outputs.
3. To maintain the background of the source image, the article utilizes a cycle-consistency objective. This involves iteratively optimizing source and target latent variables, enhancing the feasibility of the method.

**Weaknesses:**

I find this method to be intuitive, but it appears to lack enough technical innovation, as similar concepts have been previously mentioned in prior works. My primary concerns are related to the experimental aspects:

1. The authors should also conduct experiments on some of the datasets or images provided in their previous work.

2. The quantitative experiments in the study appear to be insufficient. Since the authors have collected a dataset, it would be better for them to report average metrics on this dataset.

3. There is a shortage of comparison with other methods. Given the wide attention in this field, it would be beneficial to compare this approach with more recent works. It is also better to include some fine-tuning-based methods (like SINE, Text-Inversion)  to provide a more comprehensive evaluation.

4. The running costs, such as time and GPU resource consumption, should be reported and compared to help readers understand the resource requirements when using this method.

5. The authors have not listed the limitations of their method. As this approach is positioned as a general method, it is essential to clarify whether it supports general scenarios, like the removal or addition of specific elements in the target image, to better inform users about its applicability.

6. When comparing "Prompt-to-prompt," it seems that the authors have not adopted a strategy that specifically considers the background region. This might impact the accuracy of the experimental results.

**Questions:**

See Weaknesses.

---

> ### Author Response · Authors · 2023-11-14
> **Rebuttal by Authors**
>
> We truly thank you for your constructive comments and below are our responses to the main questions.
>
>
> Q1. Dataset for evaluation
>
>
> A1. Following the state-of-the art method [D1], we used the LAION-5B dataset to evaluate the text-driven image editing performance.
>
> Q2. Measurement of metrics using the entire images
>
> A2. As we mentioned in the main paper, we simply selected about 250 images for all tasks from the LAION-5B dataset based on the CLIP similarity for quantitative experiments since most of the remaining images can be irrelevant to the given tasks.
>
> Q3.Comparison with SINE and Null-text Inversion
>
>
> A3. Since Null-text Inversion is orthogonal to our method, Null-text Inversion can be incorporated into our framework to further enhance the performance. We emphasize that Null-text Inversion is originally proposed to encourage the reconstructed image to align with the source image.  In case of SINE, it requires an additional fine-tuning process on the pretrained Stable Diffusion different from the proposed method. For fair comparisons, we focused on performing comparisons with training-free methods.
>
> Q4. Limitations
>
>
> A4. As we mentioned in the main paper, the proposed method can generate harmful or misleading samples due to the pre-trained model. For example, the pre-trained network can generate realistic samples that can potentially violate the privacy.
>
> Q5.Strategy about preserving the background region
>
> A5. Our triplet-based distance objective is effective to maintain the background region. Also, the cycle-consistency objective encourages the target images to preserve the structural and background information of the source images.
>
> Reference
>
> [D1] G. Parmar et al., Zero-Shot Image-to-Image Translation, SIGGRAPH 2023.

---

### Official Review · Reviewer_GXjj · 2023-10-31

**Soundness:** 2 fair
**Presentation:** 3 good
**Contribution:** 3 good
**Rating:** 6
**Confidence:** 5

**Summary:**

The paper  presents a training-free approach for text-driven image-to-image translation using a pretrained text-to-image diffusion model.

**Strengths:**

1.The paper introduces a new guidance objective term, which combines maximizing similarity to the target prompt (based on the CLIP score) and minimizing the distance to the source latent variables.

2.Unlike many existing methods based on diffusion models, the paper leverages a cycle-consistency objective to preserve the background of the source image.

**Weaknesses:**

1. The time consumption of the proposed method compared to other methods should be given.


2. Comparable works such as "Negative-prompt Inversion" and "Null-text Inversion for Editing Real Images using Guided Diffusion Models" demonstrate robust image reconstruction and content editing capabilities while preserving the original background. These works also support flexible target category transformations. A comprehensive comparison with these similar works could further support the paper's novelty and performance in relation to existing solutions.

3. A user study is encouraged to be carried out.

**Questions:**

see above

---

> ### Author Response · Authors · 2023-11-14
> **Rebuttal by Authors**
>
> We truly thank you for your constructive comments and below are our responses to the main questions.
>
> Q1. Comparison with Negative-prompt Inversion and Null-text Inversion
>
>
> A1. Since Negative-prompt Inversion and Null-text Inversion are orthogonal to our method, they can be incorporated into our framework to further enhance the performance. We emphasize that Negative-prompt Inversion and Null-text Inversion are originally proposed to encourage the reconstructed image to align with the source image
>
>
> Q2. User study
>
>
> A2. We acknowledged the importance of the user study, however, please understand the difficulty to perform it in the rebuttal period.

---

### Official Review · Reviewer_q62i · 2023-11-01

**Soundness:** 2 fair
**Presentation:** 2 fair
**Contribution:** 2 fair
**Rating:** 3
**Confidence:** 4

**Summary:**

The paper introduces a unique method for free text-driven image editing by utilizing pre-trained text-to-image diffusion models. Central to this approach is a new guidance objective term, which maximizes similarity to the target prompt (as opposed to the source prompt) based on the CLIP score. In tandem, it minimizes the distance to the source latent variables. Additionally, the authors incorporate a cycle consistency objective to retain the background details.

**Strengths:**

- **Simplicity & Effectiveness**: The proposed method is both straightforward and seemingly efficacious, as evidenced by the results presented in the paper.

**Weaknesses:**

- **Evaluation Methods**: The evaluation could be more robust. The prompts used for evaluation are closely related to the original noun, reducing diversity and potentially biasing results.
- **Aspect Ratio Concerns**: The samples used for evaluation have been altered from their original aspect ratios. This could inadvertently disadvantage competing methods.
- **Comparison Choices**: The results from prompt-to-prompt evaluations seem to perform well on generated images rather than inverted real ones. The absence of a comparison with Null-text-inversion, which might be a more apt benchmark, raises questions.
- **Efficiency Metrics**: The paper would be more informative with a runtime efficiency comparison against other methods.

**Questions:**

None

---

> ### Author Response · Authors · 2023-11-14
> **Rebuttal by Authors**
>
> We truly thank you for your constructive comments and below are our responses to the main questions.
>
>
> Q1. Evaluation protocol
>
> A1. We tried to follow the experiment protocol of the state-of-the-art methods [B1, B2] in terms of the text-drivn image-to-image translation tasks.
>
> Q2. Aspect ratio concerns
>
>
> A2. We are sorry for confusing you. We used the original data for all comparison methods including the proposed algorithm by keeping their original aspect ratios, but we cropped them only for the visualizations to clearly show the presented images given by the comparison algorithms.
>
> Q3. Comparison with Null-text Inversion
>
>
>  A3. Since Null-text Inversion is orthogonal to our method, Null-text Inversion can be incorporated into our framework to further enhance the performance. We emphasize that Null-text Inversion is originally proposed to encourage the reconstructed image to align with the source image.
>
>
>
> Reference
>
> [B1] A. Hertz et al., Prompt-to-Prompt Image Editing with Cross-Attention Control, ICLR 2023.
>
>
>
> [B2] G. Parmar et al., Zero-Shot Image-to-Image Translation, SIGGRAPH 2023.

---

### Official Review · Reviewer_Cs1x · 2023-11-01

**Soundness:** 2 fair
**Presentation:** 3 good
**Contribution:** 2 fair
**Rating:** 3
**Confidence:** 4

**Summary:**

The paper proposes a method for editing without training using additional cycle-consistency and triplet-based distance guidance. The triplet-based distance ensures that source and target images at the same time step are mapped closer together than those at different time steps, in addition to using a general feature similarity-based distance. The cycle-consistency objective is employed to ensure that two images, one with guide in the forward process and the other with guide in the backward process, produce identical results.

**Strengths:**

The method produces better structure-preserved editing results. Also, compared to other training-free algorithms the proposed method achieves better quantitative results.

**Weaknesses:**

Cycle-constistency may overly fix the structure and may make the result unnatural with object with different structure. Also, the argument that different time-step target images should be farther apart than same time-step source and target images seems to lack sufficient justification. And there is no comparison with papers such as null-text inversion.

**Questions:**

It seems there are only subtle differences with naive distance and the triplet distance guidance results. Is there more basis for the triplet loss that makes different time-step images of the same image distant from each other?
Also, how does the performance compare to recent papers such as null-text inversion and similar approaches?

---

> ### Author Response · Authors · 2023-11-14
> **Rebuttal by Authors**
>
> We truly thank you for your constructive comments and below are our responses to the main questions.
>
>
> Q1. Basis for the triplet loss that makes different time-step images of the same image distant from each other
>
>
> A1. Since the reverse process destroys the structure of the source image, the distance between $F( \bar{\mathbf{x}} _t^{\text{tgt}} )$ and $F ( \mathbf{x} _t^{\text{src}} ) $ should be relatively closer compared to the distance between $F( \bar{\mathbf{x}} _t^{\text{tgt}} )$ and $F ( \bar{\mathbf{x}} _{t+1}^{\text{tgt}} )$ to preserve the structure or background in the source image.
>
>
> Q2. Comparison with Null-Text Inversion
>
>
> A2. Since Null-text Inversion is orthogonal to our method, Null-text Inversion can be incorporated into our framework to further enhance the performance. We emphasize that Null-text Inversion is originally proposed to encourage the reconstructed image to align with the source image.